# Neural Complexity Measures

**Yoonho Lee**[1], **Juho Lee**[1,2], **Sung Ju Hwang**[1,2], **Eunho Yang**[1,2], **Seungjin Choi**[3]

AITRICS[1], Seoul, South Korea, KAIST[2], Daejeon, South Korea, BARO AI[3], Seoul, South Korea
eddy@aitrics.com

## Abstract

While various complexity measures for deep neural networks exist, specifying an appropriate measure capable of predicting and explaining generalization in deep networks has proven challenging. We propose *Neural Complexity* (NC), a meta-learning framework for predicting generalization. Our model learns a scalar complexity measure through interactions with many heterogeneous tasks in a data-driven way. The trained NC model can be added to the standard training loss to regularize any task learner in a standard supervised learning scenario. We contrast NC's approach against existing manually-designed complexity measures and other meta-learning models, and we validate NC's performance on multiple regression and classification tasks.

## 1 Introduction

Deep neural networks have achieved excellent performance on numerous tasks, including image classification [15] and board games [27]. Although they achieve superior empirical performance, why and how these models generalize remains a mystery. Thus, understanding which properties of deep networks allow them to generalize an important problem with far-reaching potential benefits such as principled model design and safety-aware models. To explain why deep networks generalize in practice, recent works have proposed novel *complexity measures* for deep networks [11, 12, 20, 22]. Such measures quantify the complexity of the function that a neural network represents. Ideally, such complexity measures should be good predictors of the degree of generalization of a network. However, in practice, such manually-designed complexity measures have failed to capture essential properties of generalization in deep networks, such as improving with network size and worsening with label noise.

To overcome such limitations, we propose an alternative data-driven approach for constructing a complexity measure. Our model, Neural Complexity (NC), meta-learns a neural network that takes a predictor as input and outputs a scalar. Similarly to previous complexity-based generalization bounds, we provide a probabilistic bound of the true loss using NC. Our bound has very different characteristics from previous generalization bounds: it depends on both data distribution and architecture, and more importantly, becomes tighter as the NC model improves.

Experimentally, we show that a learned NC model consistently accelerates training in addition to preventing overfitting. We also show the degree to which the measure learned by NC transfers to different hypothesis classes, such as using a different network architecture, learning rate, or nonlinearity for the task learner. Compared to other recent meta-learning methods [34], the meta-learned knowledge in NC is much more stable across long learning trajectories. Finally, while most meta-learning works focus on improving performance on small tasks such as few-shot classification, we show that NC is also capable of regularizing learning in single large tasks.

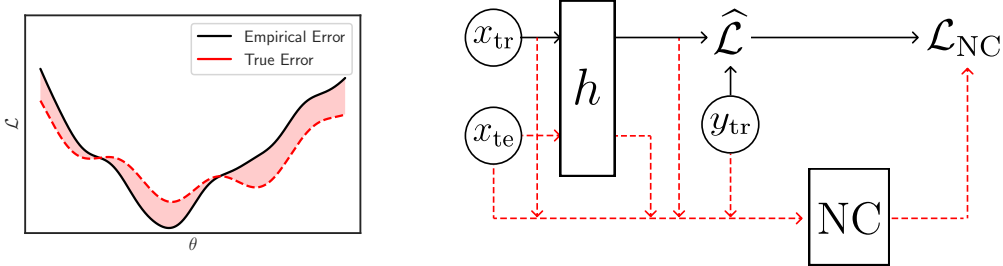

**Figure 1:** (left) The true and empirical losses are correlated but different. Neural Complexity (NC) estimates their difference (colored). (right) The training loss $\widehat{\mathcal{L}}$ is regularized (solid lines) by the output of the trained NC model (dotted lines). NC is meta-learned so that $\mathcal{L}_{NC}$ mimics the test loss.

## 2   Problem Setup

We adopt a meta-learning problem formulation in which a model (the "meta-learner") facilitates learning in new tasks using previous experience learning in other related tasks. Specifically, we assume all tasks share sample space $\mathcal{Z}$, hypothesis space $\mathcal{H}$, and loss function $\mathcal{L} : \mathcal{H} \times \mathcal{Z} \to \mathbb{R}$. Each task $T$ consists of i.i.d. sampled finite training set $S = \{z_1, \ldots, z_m\}$ from the underlying hidden distribution $D_T$ over the sample space $\mathcal{Z}$ associated with task $T$. The *true loss* $\mathcal{L}_T$ and *empirical loss* $\widehat{\mathcal{L}}_{T,S}$ for each task $T$ are respectively defined as

$$\mathcal{L}_T(h) \stackrel{\text{def}}{=} \mathbb{E}_{z \sim D_T} \left[ \mathcal{L}(h, z) \right] \quad \text{and} \quad \widehat{\mathcal{L}}_{T,S}(h) \stackrel{\text{def}}{=} \frac{1}{m} \sum_{z \in S} \mathcal{L}(h, z). \tag{1}$$

Tasks themselves are sampled i.i.d. from a distribution of tasks: $T \sim \tau$. The objective of our meta-learner is to predict the difference between the true and empirical losses, otherwise known as the *generalization gap* $G_{T,S}$:

$$G_{T,S}(h) = \mathcal{L}_T(h) - \widehat{\mathcal{L}}_{T,S}(h). \tag{2}$$

In other words, our model meta-learns a mapping $\mathcal{H} \to \mathbb{R}$ which mimics $h \mapsto G_{T,S}(h)$ by observing $\mathcal{L}_T(h)$ and $\widehat{\mathcal{L}}_{T,S}(h)$ in many different tasks that follow $T \sim \tau$.

Even in the usual single-task supervised learning setup, we can still use this problem formulation to meta-learn by constructing a set of tasks in the following way. Given one large dataset $S = \{z_1, \ldots, z_M\}$, we randomly split $S$ into disjoint training and validation sets. For each task with this random split, the task learner uses the train set to train $h$, and the meta-learner evaluates $\mathcal{L}_T$ computed with the validation set as its target. After training a meta-learner on this simulated set of tasks, we use the same model to estimate the gap $G_{T,S}$ of the full dataset $S$. This task-splitting scheme is similar to traditional cross-validation. However, instead of choosing among a few hyperparameters, we meta-learn a neural network to mimic the complex mapping $h \mapsto G_{T,S}(h)$.

## 3   Neural Complexity

We now describe Neural Complexity (NC), a meta-learning framework for predicting generalization. At the core of NC is a neural network that directly meta-learns a complexity measure through interactions with many tasks. We show how this network integrates with any standard task learner in Figure 1. We show NC's training loop in Figure 2 and also provide a detailed description in Algorithms 1 and 2.

### 3.1   Motivation: From Gap Estimate to Generalization Bound

We motivate our meta-learning objective through a simple method of extending the identity $\widehat{\mathcal{L}}_{T,S} + G_{T,S} = \mathcal{L}_T$ to a probabilistic bound of $\mathcal{L}_T$ using any estimator of the gap $G_{T,S}$.

---
**Algorithm 1** Task Learning
---
**Require:** NC, Train and test datasets
  Randomly initialize parameters $\theta$ of learner $h$
  **loop**
    Sample minibatch $X_{\text{tr}}, X_{\text{te}}, Y_{\text{tr}}$
    $\mathcal{L}_{\text{reg}} \leftarrow \widehat{\mathcal{L}}_{T,S}(h) + \lambda \cdot \text{NC}(X_{\text{tr}}, X_{\text{te}}, Y_{\text{tr}}, h(X_{\text{tr}}), h(X_{\text{te}}))$        NC-regularized task loss (4)
    $\theta \leftarrow \theta - \nabla_\theta \mathcal{L}_{\text{reg}}$        Gradient step
  **end loop**
  $G_{T,S}(h) \leftarrow \mathcal{L}_T(h) - \widehat{\mathcal{L}}_{T,S}(h)$        Compute gap (2)
  **return** Snapshot $H = (X_{\text{tr}}, X_{\text{te}}, Y_{\text{tr}}, h(X_{\text{tr}}), h(X_{\text{te}}), G_{T,S}(h))$        Save to memory bank
---

---
**Algorithm 2** Meta-Learning
---
**Require:** Memory bank
  Randomly initialize parameters $\phi$ of NC
  **while** not converged **do**
    Sample $X_{\text{tr}}, X_{\text{te}}, Y_{\text{tr}}, h(X_{\text{tr}}), h(X_{\text{te}}), G_{T,S}(h)$ from memory bank
    $\Delta \leftarrow G_{T,S}(h) - \text{NC}(X_{\text{tr}}, X_{\text{te}}, Y_{\text{tr}}, h(X_{\text{tr}}), h(X_{\text{te}}))$
    $\phi \leftarrow \phi - \nabla_\phi \mathcal{L}_{\text{NC}}(\Delta)$        NC's loss function (5)
  **end while**
---

**Proposition 1.** *Let $D_\mathcal{H}$ be a distribution of hypotheses, and let $f : \mathcal{Z}^m \times \mathcal{H} \to \mathbb{R}$ be any function of the training set and hypothesis. Let $D_\Delta$ denote the distribution of $G_{T,S}(h) - f(S, h)$ where $h \sim D_\mathcal{H}$, and let $\Delta_1, \ldots, \Delta_n$ be i.i.d. copies of $D_\Delta$. The following holds for all $\epsilon > 0$:*

$$\mathbb{P}\left[ \left| \mathcal{L}_T(h) - \widehat{\mathcal{L}}_{T,S}(h) \right| \leq f(S, h) + \epsilon \right] \geq 1 - \frac{|\{i | \Delta_i > \epsilon\}|}{n} - 2\sqrt{\frac{\log \frac{2}{\delta}}{2n}}. \tag{3}$$

We defer the proof to the supplementary material. First, note that the role of $f$ in this bound mirrors that of complexity measures in previous generalization bounds. Since we can compute $f$ given $S$ and $h$, we can restate Proposition 1 as stating that the regularized loss $\widehat{\mathcal{L}}_{T,S}(h) + f(S, h)$ differs from $\mathcal{L}_T$ by at most $\epsilon$ (with the given probability). Furthermore, making $f$ more accurately predict $G_{T,S}$ tightens this bound by decreasing the $\frac{|\{i | \Delta_i > \epsilon\}|}{n}$ term.

Taking motivation from this result, our NC meta-learns such a function $f$ by regressing towards the gap $G_{T,S}$. Rather than designing a measure of complexity that yields a tight generalization bound, NC directly learns such a measure in a data-driven way by posing the tightening of the bound as an optimization problem.

## 3.2 Training

We first illustrate NC's training loop. Recall from Section 3 that we consider a meta-learning setup consisting of a set of related but different tasks. Given a task $T$ with dataset $S$, the task learner minimizes the following regularized loss using stochastic gradient descent:

$$\mathcal{L}_{\text{reg}}(h) = \widehat{\mathcal{L}}_{T,S}(h) + \lambda \cdot \text{NC}(h). \tag{4}$$

We set $\lambda = 0$ at initialization, and use a linear schedule where $\lambda = 1$ after a certain number of episodes. We calculate the two terms in $\mathcal{L}_{\text{reg}}(h)$ using minibatches, just as in regular supervised learning with neural networks. Further note that one can use NC regularization alongside any of the usual tricks for training, such as data augmentation or batch normalization.

The objective of NC is to estimate the difference between $\mathcal{L}_T$ and $\widehat{\mathcal{L}}_{T,S}$ for any of the hypotheses $h_0^T, \ldots, h_N^T$, for any task $T \sim \tau$. NC is a permutation-invariant neural network that takes features of the function $h$ and minibatches of the data as input and outputs a scalar. We train NC using the Huber loss [9] with target $G_{T,S}$:

$$\mathcal{L}_{\text{NC}}(\Delta) = \begin{cases} \frac{1}{2}\Delta^2 & \text{for } \Delta \leq 1 \\ |\Delta| - \frac{1}{2} & \text{otherwise} \end{cases}, \tag{5}$$

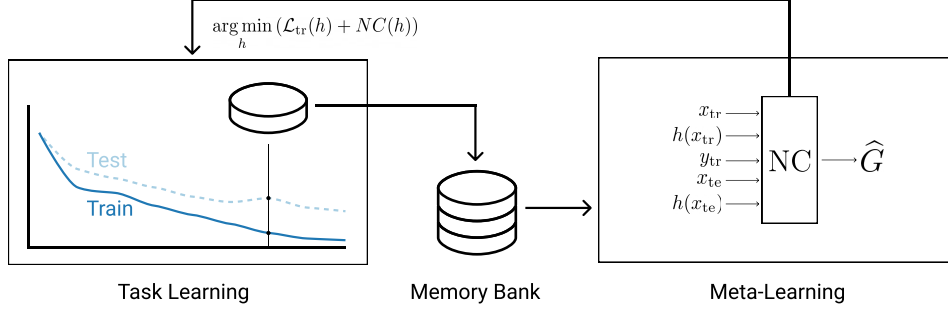

**Figure 2:** NC regularizes task learning. We store snapshots of task learning in a *memory bank*, from which we uniformly sample batches to train NC.

where $\Delta = G_{T,S}(h) - \text{NC}(h)$. We found that the Huber loss was more stable than the standard MSE loss, likely because the scale of $G_{T,S}$ can vary widely depending on $h$.

### 3.3 Architecture

We now describe the architecture of NC used in our experiments. We have mentioned in Proposition 1 and Equation 4 that NC takes a representation of the function $h$ as input. We accomplish this by passing both data $x$ and predictions $h(x)$ to NC. In terms of the function $h$, the tuple $(x, h(x))$ can be written as $\delta_x[(\text{id}, h)]$ where $\delta_x$ is the evaluation functional at $x$ and id is the identity function. This representation of $h$ captures the behavior of $h$ at the datapoints considered during training and evaluation. In our experiments, this structure sufficed for extracting relevant features of $h$.

**Regression**   We first describe NC when each task $T$ is a regression task with vector data ($x \in \mathbb{R}^D$). Let $X_{\text{tr}} \in \mathbb{R}^{m \times D}$, $X_{\text{te}} \in \mathbb{R}^{m' \times D}$, $Y_{\text{tr}} \in \mathbb{R}^{m \times 1}$ denote train data, test data, and train labels, respectively. The learner's hypothesis $h$ produces outputs $h(X_{\text{tr}}) \in \mathbb{R}^{m \times 1}$ and $h(X_{\text{te}}) \in \mathbb{R}^{m' \times 1}$ for train and test data. NC first embeds all data with a shared encoding network $f_{\text{enc}}$, which is an MLP that operates row-wise on these matrices:

$$f_{\text{enc}}(X_{\text{tr}}) = e_{\text{tr}} \in \mathbb{R}^{m \times d}, \quad f_{\text{enc}}(X_{\text{te}}) = e_{\text{te}} \in \mathbb{R}^{m' \times d}. \tag{6}$$

These embeddings are fed into a multi-head attention layer [33] where queries, keys, and values are $Q = e_{\text{te}}$, $K = e_{\text{tr}}$, $V = [e_{\text{tr}}, y_{\text{tr}}] (\in \mathbb{R}^{m \times (d+1)})$, respectively. The output of this attention layer is a set of $m'$ items, each corresponding to a test datapoint:

$$f_{\text{att}}(Q, K, V) = e_{\text{att}} \in \mathbb{R}^{m' \times d}. \tag{7}$$

Finally, these embeddings are passed through a decoding MLP network and averaged:

$$\text{NC}(X_{\text{tr}}, X_{\text{te}}, Y_{\text{tr}}, h(X_{\text{tr}}), h(X_{\text{te}})) = \frac{1}{m'} \sum_{i=1}^{m'} f_{\text{dec}}(e_{\text{att}})_i \in \mathbb{R}. \tag{8}$$

Note that NC is permutation invariant because it is consists of permutation invariant components. This property is essential since NC's objective is also invariant with respect to permutation of the input dataset.

**Classification**   The architecture of NC for classification tasks is identical to that of regression, except for the following additional interaction layer used to compute $V$. Representing labels as one-hot vectors in a classification task with $c$ classes gives $Y_{\text{tr}} \in \mathbb{R}^{m' \times c}$. Instead of concatenating $e_{\text{tr}}$ and $Y_{\text{tr}}$ as in (7), we use a bilinear layer to produce $V$:

$$V = \mathbb{W}(e_{\text{tr}}, [Y_{\text{tr}}, \mathbf{1}, \mathcal{L}(X_{\text{tr}})]) \in \mathbb{R}^{m' \times d} \quad (\mathbb{W} \in \mathbb{R}^{d \times d \times (c+2)}). \tag{9}$$

Note that we concatenate a vector of ones and the train loss to $Y_{\text{tr}}$ before passing into the bilinear layer: this vector acts like a residual connection for the embedding $e_{\text{tr}}$, allowing its information

to freely flow to the next layer. See Section 5 for an ablation study on each of our architectural choices. The bilinear layer (Equation 9) generalizes the interaction layer proposed in [36]: while they explicitly choose a subnetwork to use according to class, (9) implicitly multiplies 0 to all but one of the $c$ weights in each slice of the last dimension. Additionally, to scale NC up to high-dimensional image data such as the CIFAR dataset, we use a convolutional neural network for the encoder $f_{\mathrm{enc}}$.

Because training runs (4) are time-consuming for large networks $h$, we use a *memory bank* to store and re-use the information necessary for the meta-learning loss (5). Specifically, we store tuples $(X_{\mathrm{tr}}, X_{\mathrm{te}}, Y_{\mathrm{tr}}, h(X_{\mathrm{tr}}), h(X_{\mathrm{te}}))$ along with the observed gap $\widehat{\mathcal{L}}_{T,S}(h) - \mathcal{L}_T(h)$. This memory bank has manageable memory cost because we can store only the indices for $X_{\mathrm{tr}}, X_{\mathrm{te}}$, and the other tensors have low dimensions. We randomly sample minibatches of such tuples to train NC with the meta-learning loss (5). Figure 2 shows how the memory bank interacts with NC.

## 3.4 Interpretations

We provide several different interpretations the NC framework.

**Causality in Generalization** As noted in [11], the correlation between a complexity measure and generalization does not directly imply a causal relationship. Since regularizing the complexity measure implicitly assumes that reducing the measure will cause the model to generalize, this lack of a causal connection can be problematic. NC's framework provides an alternative way around this issue: because we keep using NC as a regularizer, it continually gets feedback on whether its predictions have caused the task learner to generalize.

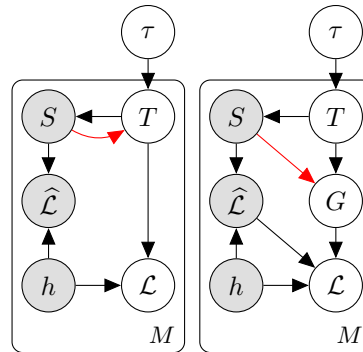

**Figure 3:** Graphical models corresponding to (left) Neural Processes and (right) NC. Observed nodes are shaded and red arrows denote amortized inference.

**Meta-learned Complexity Measure** As mentioned in Section 1, many recent works attempt to understand generalization in deep networks by proposing novel complexity measures. Such measures are designed to correlate well with generalization while being directly computable for any given set of parameters. NC can be seen as a meta-learned complexity measure, and its target (2) is the generalization gap. Instead of hand-designing an appropriate complexity measure, NC meta-learns it by regressing towards observed degrees of generalization.

**Optimal Regularizer** A standard approach to generalization is to augment the empirical loss $\widehat{\mathcal{L}}_{T,S}$ by adding a regularization term $\lambda$: $\mathcal{L}_{\mathrm{reg}} = \widehat{\mathcal{L}}_{T,S}(h) + \lambda(h)$. Since the purpose of $\lambda$ is to make the regularized loss $\mathcal{L}_{\mathrm{reg}}$ close to the true loss $\mathcal{L}_T$, we argue that the *optimal regularizer* for task $T$ is the function that makes $\mathcal{L}_{\mathrm{reg}}(h) = \mathcal{L}_T(h)$ for all $h$. This unique "optimal regularizer" is exactly $G_{T,S}$; therefore, NC can be seen as a learned approximation to this optimal regularizer.

**Neural Processes and Sufficient Statistics of True Loss** We contrast the graphical models of NC with the Neural Process (NP) [7] in Figure 3. Both approaches involve a single meta-learner which observes multiple tasks to achieve low test loss. The two approaches infer different sufficient statistics for the true loss $\mathcal{L}_T$. NP infers the data distribution of $T$ and NC infers the gap $G$. While both $G$ and the data distribution are sufficient for reconstructing $\mathcal{L}_T$, $G$ has much lower dimension: $G(h) \in \mathbb{R}$, whereas $T$ is a complex distribution over $\mathcal{X} \times \mathcal{Y} = \mathbb{R}^{d_X + d_Y}$.

**Actor-Critic** Generalizing to unseen test data can be seen as a reinforcement learning environment with known dynamics: the observations are train data and train loss, and the selection of hypothesis $h$ is the action. The objective is to maximize the return, which is $-\mathcal{L}_T$. Within this interpretation, our approach is an actor-critic method where NC takes the value network's role.

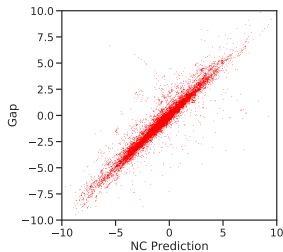

**Figure 4:** NC predictions and true gap values.

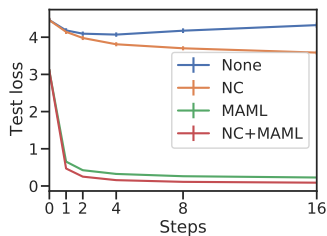

**Figure 5:** Test loss of combinations of NC and MAML.

| Method | Accuracy |
|---|---|
| Full | **92.93** |
| - Huber loss | 89.74 |
| - Bias | 84.30 |
| - Loss conditioning | 72.09 |
| - Bilinear layer | 23.69 |

**Figure 6:** Ablation study; we sequentially removed each architectural component.

## 4   Related Works

**Complexity Measures for Deep Networks**   The question of why deep networks generalize despite being over-parameterized has been the focus of many recent works. Building on traditional generalization theory [32, 21], such works have adapted PAC-Bayes bounds [5, 39] and norm-based bounds [24, 2] to deep networks. Other works have proposed measures that empirically correlate with generalization [12, 20, 22]. This work proposes an alternative approach to the problem of explaining generalization. While these previous works rely on human-designed measures of complexity, NC learns such a measure in a data-driven way, allowing it to learn to reflect the complex interaction between the function $h$ and the data distribution.

**Predicting Generalization**   A few recent works have proposed to predict generalization using function approximation. [10] learns linear regression coefficients to predict the generalization gap, and [37] extends their model by using a small neural network instead of a linear model. [31] proposes to predict neural network accuracy from various weight features. These approaches are all *correlational* analyses of generalization. As discussed in Section 3.4, NC's framework allows for the discovery of causal factors of generalization because of the back-and-forth interaction between task learning and predicting generalization.

**Meta-Learning**   Our method falls within the framework of meta-learning [30, 26], in which a model learns useful information about the learning process itself through interactions with a set of different but related tasks. Recent methods formulate the meta-learning problem as learning optimizers [25], data embeddings [28], initial parameters [6], or parameter priors [13]. A key difference is that NC is learner-agnostic: we can use an NC model trained on one class of task learners to regularize other task learners (e.g., different architecture, activation, optimizer). Additionally, using NC's output as a regularization loss makes it more stable in long training runs than previous meta-learning algorithms.

MetaReg [1] proposes to meta-learn a weighted $L_1$ regularizer. It is probably the most similar method because they also learn a regularizer in a meta-learning setup. However, their regularizer does not transfer to different network architectures because it operates in parameter space. In contrast, NC learns a complexity measure in function space, allowing it to generalize to different architectures and randomly initialized networks.

## 5   Experiments

### 5.1   Sinusoid Regression

To illustrate the basic properties of NC, we begin with a toy sinusoid regression problem introduced in [6]. Each task is a sine function $x \mapsto A\sin(x + b)$ where $x, A, b$ are uniformly sampled from $[-5, 5]$, $[0.1, 5]$ and $[0, \pi]$, respectively. We consider 10-shot learning and use the mean squared error as the loss function. We measure test losses using a held-out test set of 15 datapoints. During meta-training, the layer size, activation, number of layers, learning rate, and number of steps were all fixed to (40, ReLU, 2, 0.01, 16), respectively.

**NC-Gap Fit**   In Figure 4, we compare the predictions of a trained NC model with the generalization gap $G_{T,S}$ for many batches. These two values are strongly correlated ($R^2 = 0.9589$), indicating that NC is indeed capable of predicting the gap based on the complexity of the learner's hypothesis $h$.

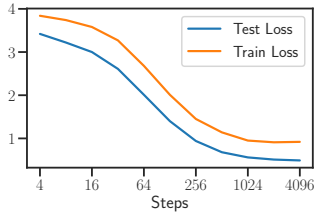

**Figure 7:** Train and test loss curves of NC regularization.

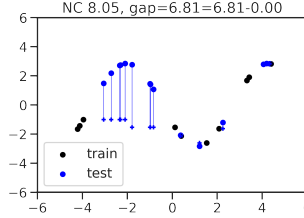

**Figure 8:** Nearest neighbor regression. Circles and plus signs represent targets and predictions.

|  | SVHN | CIFAR |
|---|---|---|
| Baseline | 93.23 | 79.76 |
| size $\times 2$ | 93.59 | 79.64 |
| NC (ours) | **93.83** | **81.15** |
| size $\times 4$ | 93.88 | 80.47 |

**Figure 9:** Comparison to networks with more capacity.

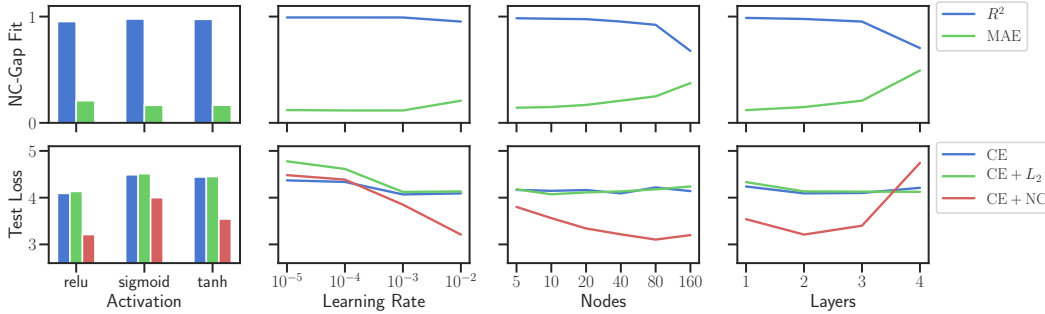

**Figure 10:** Evaluation of Out-of-distribution task learners. The x axis shows the altered learner hyperparameter. (top) NC-gap fit statistics and (bottom) test loss after learning with NC and baselines.

**VS Other Regularizers**   We compared NC against various other regularization methods. We report these results in the appendix due to space issues. NC performs all other methods by a large margin because learners tend to overfit very quickly in this few-shot regression problem.

**Integration with MAML**   We investigated whether NC can integrate with MAML [6], an alternative meta-learning approach. We first note that these two methods solve two very different problems: NC aims to regularize any randomly initialized network while MAML simply finds one set of initial parameters from which learning occurs quickly. We first trained a MAML model and then trained NC using snapshots obtained from MAML trajectories. Figure 5 shows that NC successfully reduces the final test loss for both settings: with and without MAML initializations. These results indicate that the regularization effect of NC is orthogonal to that of MAML, and that future improvements in either direction can benefit the other.

**Learning Curve**   We show the train and test loss curves of a task learner regularized by NC in Figure 7. The test loss is lower than the train loss throughout training, which is a trend that we observed in all experimental settings we considered. In other words, the estimate of NC is a precise enough surrogate for $G_{T,S}$ that minimizing it results in negative $G_{T,S}$.

## 5.2   Out-of-distribution Task Learners

**Visualization of Simple Learners**   We observed the behavior of NC when given hypotheses from closed-form learners with very distinct properties. We consider a nearest neighbor learner. We show $(X_{\mathrm{tr}}, X_{\mathrm{te}}, Y_{\mathrm{tr}}, Y_{\mathrm{te}}, h(X_{\mathrm{tr}}), h(X_{\mathrm{te}}))$ and gap values along with NC predictions in Figure 8. Note that while we show test targets $Y_{\mathrm{te}}$ in the figure, NC does not observe them. Even though the shown predictive function $h$ perfectly fits the training data, NC penalizes the function because it does not have a sinusoidal shape.

**Changing NN Learner Hyperparameters**   We evaluated how well NC can generalize to other task learners on the sinusoid regression task in Section 5.1. We measured performance while alternating four different learning algorithm hyperparameters: activation, learning rate, nodes per layer, and the number of layers. We used the same NC model which was trained using only (relu, $10^{-2}$, 40, 2) for these hyperparameters. In Figure 10, we measure how well NC fits the $G_{T,S}$ through their $R^2$ statistic and their mean absolute error (MAE). NC's predictions are accurate even when the learners are

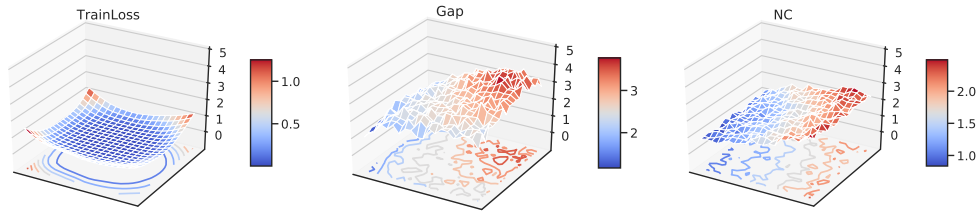

**Figure 11:** Visualization of loss surfaces. Best viewed zoomed in.

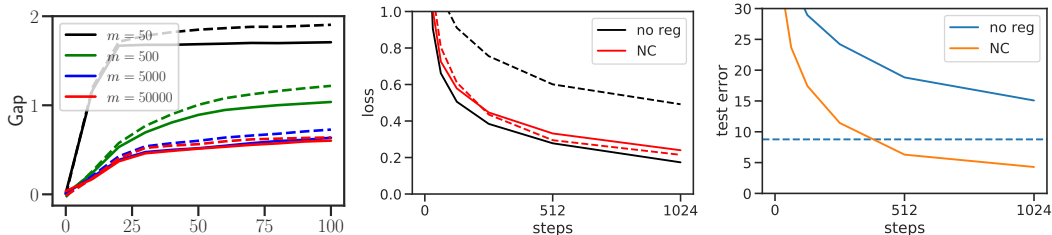

**Figure 12:** Single-task regularization on the KMNIST dataset. Left: NC estimates (solid lines) and gap values (dashed lines) when training with datasets of different size. Center: Learning curve of train (solid lines) and test (dashed lines) losses with and without NC. Right: Learning curve of test error with and without NC. The dashed horizontal line represents baseline performance after convergence.

changed, only degrading when the learner is significantly more expressive (160 nodes and 4 layers). Figure 10 additionally reports the test losses of NC regularization compared to the cross-entropy loss and $L_2$ regularization. NC regularization shows consistent improvements except for when the network architecture was too different (4 layers). This experiment demonstrates that NC's complexity measure captures the properties of $h$ itself, regardless of its specific parameterization. We emphasize that such a transfer between different task learners is not possible with other meta-learning approaches. [6, 28, 7].

### 5.3 Few-shot Image Classification

**Ablation Study** To validate our architectural choices for NC, we performed an ablation experiment using a 10-way 1-shot classification task on the Omniglot [16] dataset. Results in Figure 6 show the performance of several architectures for NC. First note that removing Huber loss and using MSE loss degrades performance, likely due to large gradients when the difference between NC's prediction and $G_{T,S}$ is large. Furthermore, removing any of the additional components for the classification model (bias, train loss, bilinear layer) reduces accuracy, with the bilinear layer being the most critical for performance.

**Loss Surface Visualization** We use the filter-wise normalization technique introduced in [19] to visualize the loss surfaces of a learner at convergence. Figure 11 shows that the train loss is at a stable local minimum, but the generalization gap can decrease further by moving in a specific direction. Because NC correctly captures the trend of the gap, minimizing the NC-regularized loss would move the learner to a region with lower test loss.

### 5.4 Single Image Classification Tasks

Finally, we evaluate NC on single tasks, following our protocol outlined in Section 3 of constructing a large number of sub-tasks using only the train split and then evaluating on the test set. We consider five different datasets: three MNIST variants (MNIST [17], FMNIST [35], KMNIST [3]), for which the learner was a 1-layer MLP with 500 units, and SVHN [23] along with CIFAR-10 [14] , for which we used the ResNet-18 [8] network. To isolate the effect of the regularizers, we do not use image augmentation or manual learning rate scheduling. Due to space constraints, we describe detailed hyperparameters and NC architectures in the appendix.

|  | MNIST | FMNIST | KMNIST | SVHN | CIFAR-10 |
|---|---|---|---|---|---|
| Cross-Entropy | $98.31_{\pm0.12}$ | $88.21_{\pm0.05}$ | $91.12_{\pm0.08}$ | $93.23_{\pm0.44}$ | $79.76_{\pm0.34}$ |
| $L_2$ Regularization | $98.36_{\pm0.06}$ | $88.46_{\pm0.17}$ | $91.31_{\pm0.05}$ | $94.06_{\pm0.44}$ | $79.84_{\pm0.75}$ |
| Label Smoothing [29] | $98.57_{\pm0.11}$ | $89.15_{\pm0.40}$ | $91.40_{\pm0.05}$ | $94.70_{\pm0.38}$ | $80.45_{\pm0.44}$ |
| Mixup [38] | $97.80_{\pm0.27}$ | $89.50_{\pm0.16}$ | $91.10_{\pm0.17}$ | $\mathbf{94.88}_{\pm0.24}$ | $80.92_{\pm0.47}$ |
| NC Regularization | $\mathbf{99.03}_{\pm0.07}$ | $\mathbf{89.74}_{\pm0.17}$ | $\mathbf{96.30}_{\pm0.07}$ | $93.83_{\pm0.18}$ | $\mathbf{81.15}_{\pm0.36}$ |

**Table 1:** Mean test accuracies and $95\%$ confidence intervals of each method on 5 runs.

**Network Size** To test whether the gains from NC are simply from the additional parameters, we compared against larger networks in Section 5.2. We constructed networks with $2\times, 4\times$ capacity by jointly training multiple networks with a pooling layer. These results show that NC is much more effective compared to simply using a larger model, and is in fact outputting a useful approximation to $G_{T,S}$.

**Effect of Dataset Size** We investigated whether NC can capture the effect of dataset size $m$ on overfitting. Using an unregularized learner on the KMNIST dataset, we measured how the gap and NC's estimate of it changes during task learning. The left figure of Figure 12 shows that overfitting occurs more severely with smaller datasets, and NC successfully captures this trend.

**Regularization Performance** We measure NC's effectiveness as a regularizer, comparing it to other regularization methods for classification tasks. We consider four baselines: standard cross-entropy loss, $L_2$ loss, label smoothing [29], and Mixup [38]. Results in Table 1 show that NC consistently improves test accuracy and performs similarly to modern regularization methods, even outperforming them on some tasks. We further visualize the learning curve of a NC-regularized task learner on the KMNIST dataset in the middle and right figures of Figure 12, which show that NC accelerates training in addition to improving the final accuracy.

## 6 Conclusion

We proposed Neural Complexity (NC), a meta-learning framework for predicting generalization. We motivated NC through a generalization bound based on estimators of the generalization gap $G_{T,S}$. It treats generalization as a regression problem, learning the degree to which a function will generalize to unseen test data.. Notably, regularizing with NC consistently resulted in models with lower test loss than train loss. NC is capable of regularizing task learners from previously unseen architectures and hyperparameters. Additionally, our experiments demonstrate that NC can learn to learn in much larger tasks compared to previous meta-learning works, such as MNIST or CIFAR.

We see many exciting future directions for improvement within the NC framework. First, the current requirement of validation data ($X_{\text{te}}$) limits the applicability of our model, but this requirement is not intrinsic to the framework of NC. In future work, we will investigate architectures and training schemes that allow for the accurate prediction of $G_{T,S}$ from data and predictions from training data. We also think NC can be extended beyond supervised learning, for example, predicting the generalization gap in density estimation or self-supervised learning. Furthermore, we are interested in scaling this approach to the ImageNet dataset [4] since our experiments have shown that NC is much more scalable than other meta-learning methods.

## Broader Impact

Safety and reliability are important desiderata for machine learning models, and these properties are even more important given the recent success of black-box models such as deep neural networks. Our proposed approach can be applied to improve training in any regression or classification task, and our experiments demonstrate its ability to (1) predict the generalization gap and (2) improve test loss when used as a regularizer. NC's data-driven prediction of the generalization gap can serve as an approximate guarantee for safety-critical problems. Furthermore, future extensions of NC may enable previously impossible tasks since NC was particularly effective in settings where conventional learners overfitted.

## Acknowledgments and Disclosure of Funding

This work was supported by Engineering Research Center Program through the National Research Foundation of Korea (NRF) funded by the Korean Government MSIT (NRF-2018R1A5A1059921), Institute of Information & communications Technology Planning & Evaluation (IITP) grant funded by the Korea government (MSIT) (No.2019-0-00075), IITP grant funded by the Korea government(MSIT) (No.2017-0-01779, XAI) and the grant funded by 2019 IT Promotion fund (Development of AI based Precision Medicine Emergency System) of the Korea government (Ministry of Science and ICT).

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
