[Supplementary Material]

# Supplementary Material for Neural Complexity Measures

## A   Proof of Motivating Bound

We first invoke the following lemma which relates the empirical and true cumulative distribution functions of i.i.d. random variables.

**Lemma 1** (Dvoretzky–Kiefer–Wolfowitz Inequality). *Let $X_1, \ldots, X_n$ be i.i.d. random variables with cumulative distribution function (CDF) $F(\cdot)$. Denote the associated empirical CDF as $F_n(x) \triangleq \frac{1}{n} \sum_{i=1}^{n} \mathbf{1}_{\{X_i \leq x\}}$. The following inequality holds for all $x$ w.p. $\geq 1 - \delta$:*

$$|F_n(x) - F(x)| \leq \sqrt{\frac{\log \frac{2}{\delta}}{2n}}. \tag{A.1}$$

*Proof.* We omit the proof. The original theorem appears in [2] and was refined by [5]. This two-sided version appears in [4]. □

**Proposition 1.** *Let $D_\mathcal{H}$ be a distribution of hypotheses, and let $f : \mathcal{Z}^m \times \mathcal{H} \to \mathbb{R}$ be any function of the training set and hypothesis. Let $D_\Delta$ denote the distribution of $G_{T,S}(h) - f(S, h)$ where $h \sim D_\mathcal{H}$, and let $\Delta_1, \ldots, \Delta_n$ be i.i.d. copies of $D_\Delta$. The following holds for all $\epsilon > 0$:*

$$\mathbb{P}\left[\left|\mathcal{L}_T(h) - \widehat{\mathcal{L}}_{T,S}(h)\right| \leq f(S, h) + \epsilon\right] \geq 1 - \frac{|\{i | \Delta_i > \epsilon\}|}{n} - 2\sqrt{\frac{\log \frac{2}{\delta}}{2n}}. \tag{A.2}$$

*Proof.* Let $F(x), F_n(x)$ be the CDF and empirical CDF of $\Delta$, respectively.

$$\mathbb{P}\left(|\widehat{\mathcal{L}}_{T,S}(h) + \mathrm{NC}_S(h) - \mathcal{L}_T(h)| > \epsilon\right) = \mathbb{P}_{\Delta \sim p_{NC}}(|\Delta| > \epsilon) = F(\epsilon) - F(-\epsilon). \tag{A.3}$$

By Lemma 1, the following holds with probability $\geq 1 - \delta$:

$$F(\epsilon) - F(-\epsilon) \leq F_n(\epsilon) - F_n(-\epsilon) + 2\sqrt{\frac{\log \frac{2}{\delta}}{2n}} = \frac{n_\epsilon}{n} + 2\sqrt{\frac{\log \frac{2}{\delta}}{2n}}. \tag{A.4}$$

□

| Steps | 1 | 2 | 4 | 8 | 16 |
|---|---|---|---|---|---|
| No regularization | 4.17 | 4.04 | 4.05 | 4.04 | 4.05 |
| $L_1(\lambda = 10.0)$ | 4.21 | 4.26 | 4.26 | 4.25 | 4.25 |
| $L_1(\lambda = 1.0)$ | 4.08 | 4.00 | 3.98 | 4.03 | 4.13 |
| $L_1(\lambda = 0.1)$ | 4.07 | 3.98 | 3.95 | 4.01 | 4.12 |
| $L_1(\lambda = 0.01)$ | 4.08 | 3.98 | 3.95 | 4.04 | 4.16 |
| $L_2(\lambda = 10.0)$ | 4.10 | 4.11 | 4.17 | 4.22 | 4.32 |
| $L_2(\lambda = 1.0)$ | 4.08 | 3.98 | 3.94 | 3.96 | 4.00 |
| $L_2(\lambda = 0.1)$ | 4.07 | 3.98 | 4.03 | 4.03 | 4.14 |
| $L_2(\lambda = 0.01)$ | 4.08 | 3.98 | 3.96 | 4.04 | 4.16 |
| $L_{1,\infty}(\lambda = 1.0)$ | 4.08 | 4.04 | 4.07 | 4.08 | 4.08 |
| $L_{1,\infty}(\lambda = 0.1)$ | 4.07 | 3.98 | 3.95 | 4.01 | 4.12 |
| $L_{1,\infty}(\lambda = 0.01)$ | 4.08 | 3.98 | 3.96 | 4.04 | 4.16 |
| $L_{3,1.5}(\lambda = 1.0)$ | 4.07 | 4.03 | 4.11 | 4.09 | 4.07 |
| $L_{3,1.5}(\lambda = 0.1)$ | 4.08 | 3.99 | 3.95 | 4.00 | 4.08 |
| $L_{3,1.5}(\lambda = 0.01)$ | 4.07 | 3.98 | 3.95 | 4.04 | 4.15 |
| Orthogonal $(\lambda = 1.0)$ | 4.16 | 4.17 | 4.19 | 4.22 | 4.32 |
| Orthogonal $(\lambda = 0.1)$ | 4.08 | 4.00 | 3.96 | 3.99 | 4.06 |
| Orthogonal $(\lambda = 0.01)$ | 4.07 | 3.99 | 3.95 | 4.00 | 4.14 |
| Frobenius $(\lambda = 1.0)$ | 4.08 | 4.01 | 4.04 | 4.13 | 4.13 |
| Frobenius $(\lambda = 0.1)$ | 4.07 | 3.98 | 3.95 | 4.02 | 4.11 |
| Frobenius $(\lambda = 0.01)$ | 4.08 | 3.98 | 3.96 | 4.04 | 4.15 |
| Dropout $(p = 0.1)$ | 4.08 | 3.98 | 3.96 | 4.04 | 4.15 |
| Dropout $(p = 0.3)$ | 4.08 | 3.98 | 3.95 | 4.02 | 4.12 |
| Dropout $(p = 0.5)$ | 4.08 | 3.99 | 3.95 | 4.00 | 4.07 |
| Dropout $(p = 0.7)$ | 4.10 | 4.00 | 3.96 | 3.98 | 4.02 |
| Dropout $(p = 0.9)$ | 4.17 | 4.11 | 4.09 | 4.37 | NaN |
| MetaReg | 4.04 | 3.93 | 3.89 | 3.90 | 4.00 |
| Neural Complexity | **3.87** | **3.60** | **3.36** | **3.13** | **2.93** |

**Table B.1:** Test losses of various regularization methods after a certain number of steps.

# B  Additional Experiments

We evaluated the performance of the following regularizers on the sinusoid regression task: $L_1$ norm, $L_2$ norm, $L_{1,\infty}$ norm, $L_{3,1.5}$ norm, Orthogonal constraint, Frobenius norm, Dropout [7], MetaReg [1], and Neural Complexity (NC). We show performance after $\{1, 2, 4, 8, 16\}$ steps. Results in Table B.1 show that all other baselines fail to provide guidance in this task, while NC outperforms them by a large margin.

In Figure B.1, we show additional visualizations of regression tasks. This figure shows that NC successfully captures the trend of the generalization gap even in out-of-distribution hypothesis classes.

Figure B.2 shows additional experiments, where we additionally compare against stronger baselines (dropout and variational dropout).

Figure B.3 shows additional visualizations of loss surfaces, and reveals that the NC-regularized loss has similar trends to that of the test loss.

# C  Experimental Details

All experiments were ran on single GPUs (either Titan V or Titan XP) with the exception of the single-task image classification experiment, which was run on two.

These embeddings are fed into a multi-head attention layer [8] where queries, keys, and values are $Q = e_{\text{te}}$, $K = e_{\text{tr}}$, $V = [e_{\text{tr}}, y_{\text{tr}}](\in \mathbb{R}^{m \times (d+1)})$, respectively. The output of this attention layer is a

**Figure B.1:** Visualization of regression tasks. The x and y axes represent inputs and outputs of the task learners, respectively. Circles represent the targets and plus signs represent predictions. The NC model is trained with a neural network learner, and we evaluated on three different learners: 0-th order polynomial (left), nearest-neighbor (center), and neural networks (right).

**Figure B.2:** Additional experiments for out-of-distribution task learners. We additionally compare against Dropout and Variational Dropout.

**Figure B.3:** Visualization of loss surfaces. Best viewed zoomed in.

set of $m'$ items, each corresponding to a test datapoint:

$$f_{\text{att}}(Q, K, V) = e_{\text{att}} \in \mathbb{R}^{m' \times d}. \tag{C.5}$$

Finally, these embeddings are passed through a decoding MLP network and averaged:

$$\text{NC}(X_{\text{tr}}, X_{\text{te}}, Y_{\text{tr}}, h(X_{\text{tr}}), h(X_{\text{te}})) = \frac{1}{m'} \sum_{i=1}^{m'} f_{\text{dec}}(e_{\text{att}})_i \in \mathbb{R}. \tag{C.6}$$

### C.1 Sinusoid Regression

**Task Learner** The learner was a one-layer MLP network with $40$ hidden units and ReLU activations, and was trained with vanila SGD with a learning rate of $0.01$.

**NC Architecture** Datapoints $x$ are encoded using an MLP encoder with $n_{\text{enc}}$ layers, $d$-dimensional activations, and ReLU nonlinearities. The outputs of the encoder are fed into a multi-head attention layer with $d$-dimensional activations. The outputs of the multi-head attention layer are mean-pooled and fed into an MLP decoder with $n_{\text{dec}}$ layers, $d$-dimensional activations, and ReLU nonlinearities. We train NC with batch size $\text{bs}$ and the Adam optimizer with learning rate $\text{lr}$.

We considered the following range of hyperparameters: $n_{\text{enc}} \in \{1, 2, 3\}$, $d \in \{128, 256, 512, 1024\}$, $n_{\text{dec}} \in \{1, 2, 3\}$, $\text{bs} \in \{128, 256, 512, 1024\}$, $\text{lr} \in \{0.005, 0.001, 0.0005, 0.0001\}$. We tuned these hyperparameters with a random search and ultimately used $n_{\text{enc}} = 3$ $d = 1024$ $n_{\text{dec}} = 3$ $\text{bs} = 512$ $\text{lr} = 0.0005$.

### C.2 Classification

**Task Learner** The task learner was ResNet-18 [3] for the SVHN and CIFAR-10 datasets, and an MLP with one hidden layer of $500$ nodes and ReLU nonlinearities. To isolate the effect of the regularizers, we considered no data augmentation besides whitening. We train all networks with SGD with a fixed learning rate and no additional learning rate scheduling. The learning rate was $0.0001$ for ResNet-18 and $0.01$ for the MLP.

**NC Architecture** Datapoints $x$ are encoded using a shared CNN encoder. The CNN architecture was the $4$-layer convolutional net in [6] when the task learner was an MLP, and was ResNet-18 otherwise. We freeze all batch normalization layers inside NC. The outputs for only the train data is fed into a $n_{\text{enc}}$-layer MLP followed by a stack of $n_{\text{self}}$ self-attention layers, both with $d$-dimensional activations. These outputs are processed by a bilinear layer, and all outputs are fed into a multi-head attention layer with $d$-dimensional activations. The outputs of the multi-head attention layer are then fed into an MLP decoder with $n_{\text{dec}}$ layers, $d$-dimensional activations, and ReLU nonlinearities. We train NC with batch size $\text{bs}$ and the Adam optimizer with learning rate $\text{lr}$.

For the MLP learners, we considered the following range of hyperparameters: $n_{\text{enc}} \in \{1, 2, 3\}$, $n_{\text{self}} \in \{1, 2, 3\}$, $d \in \{60, 120, 240\}$, $n_{\text{dec}} \in \{1, 2, 3\}$, $\text{bs} \in \{4, 8, 16\}$, $\text{lr} \in \{0.0005\}$. We tuned these hyperparameters with a random search and ultimately used $n_{\text{enc}} = 1$, $n_{\text{self}} = 1$, $d = 120$, $n_{\text{dec}} = 3$, $\text{bs} = 16$, $\text{lr} = 0.0005$.

For the ResNet-18 learners, we considered the following range of hyperparameters: $n_{\text{enc}} \in \{1, 2, 3\}$, $n_{\text{self}} \in \{1, 2, 3\}$, $d \in \{200, 400, 800, 1600\}$, $n_{\text{dec}} \in \{1, 2, 3\}$, $\text{bs} \in \{2, 4, 8\}$, $\text{lr} \in \{0.0005\}$. We tuned these hyperparameters with a random search and ultimately used $n_{\text{enc}} = 1$, $n_{\text{self}} = 3$, $d = 400$, $n_{\text{dec}} = 3$, $\text{bs} = 4$, $\text{lr} = 0.0005$.

**Single-task Experiment Details** We provide further details about the single-task experiments. The datasets we considered had either $50000$ or $60000$ training datapoints. We constructed learning tasks from such training sets by sampling $40000$ "training" datapoints and $10000$ validation datapoints. Using such splits, we trained NC as usual. To scale to long learning trajectories, we trained NC using one process, while simultaneously adding trajectories from a separate process on a separate GPU which only ran task learners regularized by the NC model. During final evaluation, we clipped NC estimates below $-0.1$, which has the effect of ignoring NC when it is overconfident about generalization. We found that such clipping is critical for performance on long training runs.