[Reviews · NeurIPS 2020]

Review 1

Summary and Contributions: The paper proposes a data-driven complexity measure Neural Complexity (NC) for neural networks. Different from hand-designed complexity measures, NC learns to predict the generalization gap between true loss and empirical loss via meta-learning approaches. NC takes a hypothesis and training set as input, and outputs a scalar as the complexity measure. The parameters of NC are meta-trained through a pool of historical tasks. The paper further utilizes the NC module in meta-training and single training processes, by applying the output of NC into training loss as a regularization term. Experimental results demonstrated the effectiveness of NC as a regularizer.

Strengths: The paper raises the discussion of data-driven complexity measures, which is an interesting topic particularly in the meta-learning area, and proposes a novel formulation NC to this end. NC is empirically effective across multiple tasks including regression and classification.

Weaknesses: The paper posits NC as a complexity measure. Instead of characterizing the complexity of a hypothesis space, however, NC predicts the generalization gap of a particular hypothesis. It seems that NC serves more suitably as a regularization technique, and lacks theoretical justification as a complexity measure. Moreover, the training of NC involves knowledge of the true loss, in Equation (5) and the second last line of Algorithm 1. It is quite confusing how the true loss is obtained, even for training tasks following the same task distribution.

Correctness: The claims and methodology seem correct, except that the true loss in the training procedure is not justified.

Clarity: The paper is relatively clearly written. Some detailed comments about Algorithm 1: At the beginning of Algorithm 1, parameters \theta of learner h are randomly initialized. What is a parameterized learner h? Does h stands for an architecture, or a set of parameters itself? If the parameters are randomly initialized for each training task, is there a final training procedure for the model (task learners)?

Relation to Prior Work: The paper compares NC with MetaReg, a meta-learning regularization approach, in details. The discussion about the relation between NC and previous complexity measures is not sufficient.

Reproducibility: Yes

Additional Feedback: ============== update ======================== After checking the rebuttal and other comments, some of my key concerns have been resolved, and so I decide to increase my rating accordingly. In general, I think it deserves to be shared among the community. For future work, it is interesting to see more studies on its applications and theory.


Review 2

Summary and Contributions: This paper proposes a learning based neural complexity (NC) measure which takes the training data and an estimator as input and learns the degree to which the input estimator will generalize to unseen test data. The learned NC measure can then be used as an additional regularize for standard training losses. Experiments on both regression and classification tasks are conducted to verify the effectiveness. After reading the feedback and other reviewers' comments, I insist my preliminary rating.

Strengths: The paper is well written and the proposed algorithm is easy to follow. The idea of learning neural complexity measure in a data-driven manner is also interesting and shows substantial imprisonment on practical tasks.

Weaknesses: * The proposed NC measure takes the whole training and test datasets as input. I can hardly imagine how this method can be learned and applied to large scale datasets (e.g. ImageNet). Is there any solution to address the scalability issue? Otherwise, the practical contribution of this paper will be significantly reduced. * There are many missing details regarding the experiments, which make the proposed method hard to reproduce. See the Clarity section for more comments.

Correctness: The mathematical derivations seem to be correct.

Clarity: Please provide more details of the experiments. For instance, how many task learners are selected for each application in the experiments? How many data points are sampled for each task learner? How long will training NC convergence?

Relation to Prior Work: Related work is clearly discussed.

Reproducibility: No

Additional Feedback: In line 76 and 231, "... our protocol outlined in Section 3 of constructing a large number of sub-tasks ...”, I think it should be Section 2 instead.


Review 3

Summary and Contributions: The paper proposes to use a neural-network-computed scalar as the complexity measure of a learn function, neural complexity. The neural network is meta-learned and takes in the training data, training label, plus the test data and outputs a complexity value for the model. Further, the neural network uses transformer-like architecture to make itself invariant to parameterization of the function and the order of the data. The paper also demonstrates that the complexity correlates well with the true generalization gap of the models and also can be optimized to improve generalization.

Strengths: This is an extremely interesting paper. There are prior works that attempt to accomplish similar objectives but, to my knowledge, none of them does so in the function space. Computing the complexity on the function space is reasonable as an ideal property of a good complexity is invariance to reparameterization. The community at large has been focusing on analyzing the property of the parameter space (e.g. sharpness) so this functional perspective is quite refreshing. The way in which the paper incorporates data into the picture through multi-headed attention is also very clever. The empirical results are convincing (I will discuss weakness later). I am particularly intrigued by the fact that the regularized model on sinusoid regression tasks achieves a lower error than training (this however, may also be a weakness). I also enjoyed reading the interpretation section. Overall, I believe that this paper has a lot of potential and should be seen by the NeurIPS community.

Weaknesses: One of the main goals for studying complexity measures is to gain a better understanding on why a model generalizes. If a proposed complexity measure is not interpretable (in this case, a black box neural network), then it must be judged on the ground of its usefulness. To me, it seems that using neural complexity does not offer any computational advantage. First, computing the neural complexity seems extremely expensive since it requires encoding the entire training dataset, test set *every time* and this cost cannot be cached since the encoder is constantly changing (i.e. being trained). It would be interesting to see the result on a stochastic version of the algorithm and how stochastic approximation affects the complexity. Second, in a single task setting, I am not sure how much gain neural complexity has over cross-validation which offers a sharper estimate of the generalization gap in probably a shorter amount of time and can also be used to do hyper-parameter selection. If I were to be convinced that NC offers something different, then I would like to see experiments with more SOTA results on benchmark like CIFAR-10 but I doubt that’s likely to happen (although I hope to be pleasantly surprised). Lastly, if I wish to use data augmentation, how can this method be applied, would it still be useful? The paper states that to isolate the effect of the technique it does not use these techniques. This is reasonable but to get the best results data augmentation is almost indispensable and I am not sure how this technique can be combined with data augmentation and co. Furthermore, while the functional perspective is certainly nice, I am curious about how NC can be useful in the interpolation regime which is arguably one of the most mysterious aspects of deep learning. Specifically, how does NC distinguish between models that are trained to the same near perfect loss but exhibit different generalization behavior. If one were to look only at the output, they would look essentially the same. Perhaps, the signal is in the pattern of losses on individual examples rather than the average and the transformer serves as some kind nearest neighbor look up between the test set and training set? Large part of the heavy lifting is done by the data encoder. This could limit the generality of NC. I suspect that the encoder module is in some sense copying the behavior of the learner. It also has to learn a lot of data specific patterns. For example, a NC trained on CIFAR10 most likely won’t transfer to SVHN without retraining. If the goal is to predict the generalization gap, the most useful case would be on unseen data (unseen as in unavailable at the training time altogether rather than test test which is available at learning time). Then we can just use cross-validation to achieve the same goal. If the goal is to improve generalization, then there are the shortcomings listed in the previous paragraph. Of course, you can always use more datasets but I don’t think that’s very scalable. Finally, the evaluation of NC as a generalization measure could be more thorough. For example, show that NC captures changes in hyperparameters. Such experiments are currently done for the sinusoidal regression which is very simple compared to the image classification tasks. Even in the sinusoidal regression setting, the failure to capture depth is a bit worrying since it’s one of the most important parts of modern deep learning. I believe that the data of the PGDL competition at this NeurIPS would be a good fit. Minor comments: - I think it shouldn’t be called a complexity since it doesn’t provide an upper bound on the size of the hypothesis space (e.g. the output of the regression can be negative). - I am not sure why proposition 1 is needed. This is essentially a standard regression problem and therefore of course the application of such standard concentration inequality would apply. Personally I don’t think this kind of decorative math is needed since the paper itself is already very interesting. And this proposition does not make NC a complexity measure in statistical learning sense. - Delta can be negative so I think you want absolute value there. - What is the significance of the column of 1’s in the bilinear layer? I don’t think this is explained.

Correctness: The claims made in the paper are correct but may be misleading. The method is sound. Empirical evaluation could be improved. (see weakness for details)

Clarity: The paper is well written for most parts, but clarity could be improved. For figure 8, after reading the text and the caption, I still do not understand what the plots are trying to get across. For figure 10, are the gap and NC plotted the same way as the loss? If so, can the exact procedure be described? For example, how did you create the training and test set that show this property? Are they not drawn from the same distribution?

Relation to Prior Work: The paper’s discussion of prior work on meta learning seems complete but it misses several important works on predicting generalization with function approximation. [1] uses a linear regression model to predict generalization of neural networks and yields interpretable features. To my knowledge, this is the first paper where the goal is to predict the generalization gap with some form of function approximation. [2] builds on [1] with a neural network regression instead of linear. [3] also recently predicts the generalization of a model from the weights of the model using another neural network. I believe it’s important to address this paper’s connection to these works. I also find some resemblance to [4]. [1] Jiang et al., “Predicting the generalization gap in deep networks with margin distributions” , 18 [2] Yak et al., “Towards Task and Architecture-Independent Generalization Gap Predictors”, 19 [3] Unterthiner et al., “Predicting Neural Network Accuracy from Weights”, 20 [4] Bickel et al., “Discriminative Learning Under Covariate Shift”, 09

Reproducibility: Yes

Additional Feedback: --------------------------------Update--------------------------------- I thank the authors for the hardwork. After reading authors' response, I believe one of my largest concerns regarding the efficiency has been addressed. It was not clear at all in the text that they are training NC with minibatch and I trust the authors will revise the paper accordingly as they have promised in the feedbacks. The authors have also addressed my concerns regarding cross-validation, data augmentation and bilinear layer. On the other hand, a large number of smaller points have not been addressed possibly due to the time and space constraints of the feedback. This includes sign of the delta in theorem 1, necessity of theorem 1, more systematical evaluation of NC's effect, transfer between dataset and, finally, related works. Other points that authors repsonded to but did not address include behavior in interpolation regime. In the light of these changes, I would like to maintain my current evaluation. I really believe the paper has great potential but the rebuttal did not change the paper enough to warrant an increase in score. As such, I remain my original evaluation of 7. I trust that the authors will apply all the suggestions made by reviewers in final revision should the paper be accepted.


Review 4

Summary and Contributions: The paper introduces an approach to estimating the generalization gap (as measured by the difference between test and train loss) via a data-driven approach. Concretly, they propose a method to learn a function NC that takes a hypothesis and a training dataset, that can produce an estimate of what the generalization error would.

Strengths: Here's some of the things I like about this work - The method seems to be theoretically grounded with a proof sketch of how the learnt generalization gap can be close to the true generalization gap. - The meta-learning, data driven approach seems novel

Weaknesses: - Some of the notation is not explained. What is \Delta and n in Proposition-1? - Proposition-1 only presents a one-sided bound, a statement about how the regularized loss is lower-bounded by the true loss - \epsilon. Would an upper bound also be necessary here for a stronger claim? - Weak experimental comparisons: Looking at the appendix, it seems like other regularizers (especially Meta-Reg) have only been compared on the toy-sinusoid dataset. I think comparisons with Meta-Reg on MNIST and CIFAR-10 are necessary to understand the merits of NC. - Does this method scale? The NC function seems to require an entire training dataset to produce the scalar generalization gap which can be prohibitively slow.

Correctness: Yes

Clarity: The paper overall seems well-written. However, I have the following suggestions: 1. The authors can make it more clear in the introduction that the goal here is to produce a regularizer that attempts to approximate the true error. 2. The "interpretations" section could be moved after experiments, since I feel it makes the exposition more confusing. 3. In my opinion, figure-10 doesn't add a lot of insight, and can be moved to the appendix, and comparisons with other regularization techniques should be moved into the main paper (since producing a good regularizer is the main goal of this paper)

Relation to Prior Work: Yes

Reproducibility: Yes

Additional Feedback:


Review 5

Summary and Contributions: 1. The authors propose a meta-learning approach to predicting the generalization gap of a model. This gives (a) a generalization bound on the model, and, (b) optimal regularization for training a model. Significance: High 2. The authors show experimentally that the proposed approach is an effective regularizer in a few-shot setting, i.e., it prevents over-fitting. Significance: Medium 3. The authors show how the proposed method can be used to improve upon a single large task with n-fold cross validation. Significance: Medium

Strengths: S1. Significance of the contribution is high. Predicting generalization gap is a rich and novel application of meta-learning. S2. The motivation is strong, and the approach is reasonably grounded. S3. The empirical evaluation is promising but needs some baselines, more below.

Weaknesses: There is much scope for improvement in the experimental baselines as well as the quantities plotted. W1. If the claims of the paper are NC results in faster training and lesser overfitting (lines 29-30), these must be explicitly demonstrated against strong baselines for warm starting for few-shot optimization, e.g., MAML. * Figure 9 uses L2 regularization as the best baseline. This comparison is not fair in the few-shot setting because NC sees data from a lot more classes whereas L2 regularization does not. W2. In a single-task setting, it is unclear how much more expensive regular training of a network is, versus the proposed training with NC. It is also not clear if these gains are proportional to the cost, i.e., if simple learning with a larger model whose capacity is of the order of that of base model + NC model can beat the performance of NC. It would be good to compare to such a baseline. W3. Does the learnt regularizer overfit the validation set in the few-shot case? It would be interesting to compare the NC-regularized training loss (where the regularization is computed on the val set) with not only the test loss, but also the NC-regularized training loss where the regularization is computed on the test set. W4. W5. Additional ablation/hyperparameter studies suggested (a) Comparison of the performance on NC vs. size of the validation set. This one is important because I'm concerned that NC might require a large labeled validation set in order to exhibit good performance. (b) Why the exact form of eq. 9 with [y_tr, 1, L(X_tr)]? What if the ones are omitted or the loss is omitted? (c) It would help to see curves similar to those in Figure 11 for Cifar 10 (perhaps in the supplement).

Correctness: Yes

Clarity: While the description of the method is quite intuitive, there is scope for improvement, especially in presentation of the details. C1: The distinction between task training and meta-training is not clear. (a) How are these two interleaved? At what point during training are snapshots made? (b) Is each task seen multiple times or just once? (c) Does the word "steps" refer to training steps on a task or meta-training steps? E.g.: figure 5 and figure 7 and Table B.1. (d) The set up of the few-shot image classification experiments is not clear. What is the base model? How is it pre-trained or warm-started or meta-trained? C2: The description of the bilinear network for classification is unclear. Further, the NC architecture in Appendix C.2 seems to differ from this. C3: I could not find some key details. (a) In the single task setting, how many train-val splits are used for each dataset? (b) How many tasks are used for sinusoid regression? (c) During the training of NC, is each task learned to completion? If not, is there an iteration budget? (d) During the sampling of H_i in Algorithm 2, do you use the entire training and val sets from that episode? If not, do you subsample? C4. Some claims are made, which do not appear to be verified experimentally. (a) Huber loss vs MSE: the gradient magnitudes can be plotted to verify the claims of lines 85-86, 220-221. (b) Lines 164-166, 32-33: is there an experimental comparison showing that NC is more stable across long learning trajectories?

Relation to Prior Work: Yes

Reproducibility: No

Additional Feedback: Post-author-feedback --------------------- Thanks to the authors for the detailed response. Several of my questions have been answered but some are not (notably W3). I have accordingly updated my overall score. - Stronger baselines and larger models reported in the rebuttal are much appreciated. It would be good to have these comparisons for all datasets reported in the paper. ================================= I cannot reproduce this work from the paper as there are several key details which can neither be found in the main paper nor the supplement. See C1-C3. Minor comments and possible typos: * Proposition 1: the statment should hold with probability >= 1-\delta over the sampling of \Delta_1,... \Delta_n * eq. 7: is the dimension m' x (d+1)? * eq. 9: is the dimension m x d (rather than m' x d)? * Arxiv refs of some papers are shown rather than conference refs e.g., [6, 11, 12] * line 184: "performs" -> "outperforms" maybe? * What do the blue entries of Table 1 refer to? * line 110 is missing a period * Figure 6 should be a table * Figure 8 is hard to parse. What does it convey? * It might help mentioning explicitly that Section 5.2 also uses sinusoidal regression. Overall: The idea is super promising, but the paper still needs more work. I am happy to increase my score if the authors can clean up the writing and compare with strong baselines, and respond to my comments above.

[Author Response · NeurIPS 2020]

We thank all reviewers for their constructive and thorough comments. We will incorporate your many detailed comments to clean up our presentation in the next version. Most reviewers were happy with the significance and novelty of our method, while some concerns about experimental details were raised. We address your main concerns below.

| | SVHN | CIFAR |
|---|---|---|
| Baseline | 93.23 | 79.76 |
| size ×2 | 93.59 | 79.64 |
| NC (ours) | **93.83** | **81.15** |
| size ×4 | 93.88 | 80.47 |

(a) Stronger baselines for OOD experiment     (b) vs larger nets     (c) Test loss and acc trajectory

**[R1] "confusing how the true loss is obtained"** We estimate true loss $\mathcal{L}_T$ by computing the loss of a held-out set $(X_{te}, Y_{te})$, which is an unbiased estimator. We will update Alg 1 and Section 3.2 to be more clear about this point.

**[R1,3] Is NC a 'complexity measure'?** Our naming is similar to that of the Neural Process (Garnelo et al., 2018), which similarly is not a stochastic process in the strict sense. NC aims to bound the generalization gap just like traditional complexity measures such as VC-dim or Rademacher complexity. While NC does not come with theoretical guarantees, its data-driven prediction is much tighter, enabling the benefits we showed in our experiments.

**[R2,5] More details of experiments** Thank you for the suggestion. We will add more details about the task setup and training scheme to the supplementary material, and also publicly release the code used in our experiments.

**[R2,3,4] NC takes whole dataset: expensive** NC does not take entire training and test datasets as input. Instead, it takes subsets (i.e., minibatches) of the dataset at each gradient step (eq 4), just like standard SGD training. Note that the generalization gap calculated from random batches is an unbiased estimator of the true generalization gap. In the next version, we will make this critical point more apparent in Algorithm 1, eq 5, and Figure 2. We will also introduce a notation to distinguish between the entire dataset and a minibatch of it (e.g., $X_{tr}^B \subset X_{tr}, X_{te}^B \subset X_{te}, \ldots$).

**[R3] "in a single task setting, not sure how much gain NC has over cross-validation"** NC has much more expressive power compared to cross-validation as it actively alters the learning trajectory. Please note that we are already using cross-validation to select the hyperparameters for the baselines in Table 1.

**[R3] "how this technique can be combined with data augmentation"** We can use NC together with any sort of data augmentation, as long as we use the same augmentation scheme on all tasks.

**[R3] "interpolation regime... how does NC distinguish between models that are trained to the same near perfect loss but exhibit different generalization behavior."** We believe this is an important question. Our intuition is that NC compares the predictions for train and test data to see whether $h$ is a smooth and consistent function over the entire data distribution, or if it just sharply predicts the train data. The Transformer learns how to do this comparison.

**[R3] "PGDL competition is a good fit."** We agree, and we plan to enter PGDL with an improved version of NC.

**[R3,5] specific form of bilinear layer** The form in eq (9) was shaped through experiments. Note that our ablation study (figure 6) shows that the bias ("ones"), loss ($\mathcal{L}$), and the bilinear layer itself each positively contribute to performance. The motivation for concatenating ones was to let the data embedding $e_{tr}$ freely flow into the next layer.

**[R4] "Proposition-1 only presents a one-sided bound...stronger claim?"** In our proof in Appendix A, we actually proved the stronger two-sided version. We will edit the main text to show the two-sided bound.

**[R4] "Weak experimental comparisons... compare w/ MetaReg on MNIST and CIFAR-10"** Firstly, please note that we do compare against strong regularizers (label smoothing and mixup) on the datasets you mentioned in Table 1. MetaReg is not suitable as a baseline in our classification task because its task learners all share a "feature network", whereas we compare among methods that regularize networks trained from scratch.

**[R5] "strong baselines... e.g., MAML. Figure 9 uses L2...not fair."** First note that we showed NC's contribution is orthogonal to that of MAML in standard few-shot settings (fig 5). Fig 9 measures performance on *out-of-distribution task learners*: MAML can't be used here because initial parameters cannot be transferred to different architectures. Our additional experiments (1a) compare against stronger baselines: regular and variational Dropout (DO, vDO).

**[R5] "simple learning with a larger model... capacity similar to base model + NC model"** In table 1b, we evaluated nets with more capacity by jointly training $n$ nets with a linear layer on top. The computation requirements for NC is on the scale of $n = 2$ since most of the compute is used in the data encoder. We also evaluated $n = 4$ for reference. NC's regularization is much more effective compared to simply using a larger model.

**[R5] "concerned that NC might require a large labeled validation set in order to exhibit good performance"** This is not a practical issue for NC, because the validation set size is simply a hyperparameter that we use to split the "true training set". Note that after training the NC model, we use unlabeled $X_{te}$ to regularize task learners (Alg 1).

**[R5] "comparison showing that NC is more stable across long learning trajectories"** In 1c, we show the learning trajectory of an NC-regularized model on the KMNIST dataset. The model keeps improving for thousands of gradient steps, in contrast to other meta-learning methods like MAML which only work for a few gradient steps. The fact that NC is able to regularize learning in large single tasks (section 5.4) also demonstrates NC's stability in long trajectories.

[Meta-Review · NeurIPS 2020]

The reviews of this paper were positive overall. The authors propose a meta-learning approach to predict the generalization gap of a model. The approach yields a generalization bound on the model, and an optimal regularization to train a model. The experimental results show that the proposed approach is an effective regularizer in a few-shot setting. The authors also show how the proposed method can be used to improve upon a single large task with n-fold cross validation. The reviewers appreciated the significance of the contributions as well as the novelty of the perspective. They appreciated the clever use of multi-headed attention and the nuanced interpretation of the results. The reviewers expressed concerns, though, about baselines and comparisons in experimental evaluation and pointed out several minor issues. There were also more detailed concerns regarding cross-validation, data augmentation and bilinear layer. The authors submitted a response to the reviewers' comments, as well as confidential comments to the area chair. After reading the response, updating the reviews, and discussion, the reviewers feel that 'the idea is super promising, but the paper still needs more work' and that 'stronger baselines and larger models reported in the rebuttal are much appreciated'. We highly recommend to take the reviewers' comments and suggestions into account while preparing the final version, in particular the additional materials in the experiments and the additional clarifications in the text mentioned by the authors in the response. Accept.